# Optimizing Forecasted Activity Notifications with Reinforcement Learning

**DOI:** 10.3390/s23146510

**Published:** 2023-07-19

**Authors:** Muhammad Fikry, Sozo Inoue

**Affiliations:** 1Graduate School of Life Science and Systems Engineering, Kyushu Institute of Technology, Kitakyushu 808-0196, Japan; sozo@brain.kyutech.ac.jp; 2Department of Informatics, Universitas Malikussaleh, Aceh Utara 24355, Indonesia

**Keywords:** forecasted activity, notification system, reminder system, daily activity, reinforcement learning, Q-learning

## Abstract

In this paper, we propose the notification optimization method by providing multiple alternative times as a reminder for a forecasted activity with and without probabilistic considerations for the activity that needs to be completed and needs notification. It is important to consider various factors when sending notifications to people after obtaining the results of the forecasted activity. We should not send notifications only when we have forecasted results because future daily activities are unpredictable. Therefore, it is important to strike a balance between providing useful reminders and avoiding excessive interruptions, especially for low probabilities of forecasted activity. Our study investigates the impact of the low probability of forecasted activity and optimizes the notification time with reinforcement learning. We also show the gaps between forecasted activities that are useful for self-improvement by people for the balance of important tasks, such as tasks completed as planned and additional tasks to be completed. For evaluation, we utilize two datasets: the existing dataset and data we collected in the field with the technology we have developed. In the data collection, we have 23 activities from six participants. To evaluate the effectiveness of these approaches, we assess the percentage of positive responses, user response rate, and response duration as performance criteria. Our proposed method provides a more effective way to optimize notifications. By incorporating the probability level of activity that needs to be done and needs notification into the state, we achieve a better response rate than the baseline, with the advantage of reaching 27.15%, as well as than the other criteria, which are also improved by using probability.

## 1. Introduction

In today’s fast-paced world, it is essential to manage time efficiently to maintain a balance between our work and personal life. The use of technology can play a crucial role in managing our daily activities, and one such technology is the notification system that provides reminders for upcoming tasks. However, the effectiveness of the notification system depends on the timing of the reminders, and it can be challenging to identify the optimal time for notification, especially for activities in the future.

In their daily lives, some people frequently struggle with time management [1], such as prioritizing tasks, time for study, short breaks, cleaning the house, or meeting with friends, because if they focus too much on one or two things, they often forget other activities that must also be completed to achieve a life balance, which can also affect their health. Technologies have been developed to support their daily life, work, study, or research, such as personal web libraries [2], Tiimo [3], T-BOT and Q-BOT [3,4]. But people with very busy schedules, such as nurses, caregivers, or people with problems of cognition, loss of memory, or dementia [5] can forget the schedule that has been arranged, so a reminder system is required in this area [6,7,8,9].

A reminder system is a great way to stay organized and on top of tasks. This can aid in remembering essential information so that it is not forgotten. Through the use of reminders, it is possible to facilitate the recollection of vital information. Some reminder systems notify tasks based on a predetermined time and then send notifications according to this plan [6]. People can set their own reminder system according to their preferences, but they cannot be aware of impending activities; many things can occur outside of their schedule due to busyness [10], stress [11], perspective memory [12]. Therefore, schedule alterations can occur [13], rendering pre-set reminders ineffective. Planned activities may be affected by alterations in the schedule. In some cases, this may be normal, but in others, it will be a problem, and work interruptions may cause people to forget about upcoming obligations.

To solve this problem, instead of changing the schedule, we propose a different approach, which is to forecast future activities. Forecasting has been the subject of research in various fields, such as health informatics [14,15,16], human–computer interactions [17,18], and artificial intelligence [19,20]. The aim of these studies is to develop methods for predicting upcoming activities, events, monitoring, or accidents. For example, a forecasting system for water quality monitoring for aquaculture [21], large-scale wastewater surveillance, or managing milk production on dairy cows [22]. Several techniques have been proposed for forecasting, such as machine learning, probabilistic modeling, and rule-based systems [20].

Using forecasting, we can inform people of forthcoming activities so that they do not need to modify their schedules. This indicates that they can complete the task a little faster, a little later, or all at once. For instance, if there is a meeting at 1:00 p.m., and the user also needs to take medication at 1:00 p.m., there is no need to adjust the schedule because the duration of taking the medication is fleeting. However, sending forecasted results in a timely manner continues to be a challenge for this approach. In other studies, forecasting results are sent when forecasting results are obtained or depending on thresholds [16,21,23,24,25,26,27], while others create a separate schedule for sending forecast results [22,26,28,29,30,31]. Using an example of a weather information forecasting system for farmers, the system will send a push notification whenever the readings fall below the threshold [21,32]. In the case of daily activities, it is essential to consider user engagement as the recipient of the information, which means whether the user or the target has actually received the information or not. This is a challenge that must be resolved because if we send a notification when the predicted activity result is received, people may forget about the activity due to factors such as perspective memory symptoms; if we send a notification according to the start time of the forecasted activity, people may not necessarily like it at that time due to their busy schedules, which becomes a challenge.

In this paper, we are motivated to optimize forecasted activity notifications that consider multiple time alternatives for notifications. We have contributions that present two approaches aimed at optimizing notification timing as well as whether a forecasted activity needs to be carried out or not, and whether it is necessary to notify the user of it or not in activity recognition systems. The first approach is called FaTi. FaTi is notification optimization for forecasted activity with reinforcement learning. This approach focuses on notifying users based on the generated start time of the forecasted activity. However, instead of notifying users directly at the start time, we propose to provide alternative notification times in advance. The second approach is called FaPTi. FaPTi is notification optimization for forecasted activity with probabilistic and reinforcement learning. In this approach, alternative notification times are provided in advance. However, we calculate the probability of the activity needing to be completed and needing a notification before optimizing the time with reinforcement learning. By considering these factors, we are working to further enhance the effectiveness of the notification system. Both approaches offer strategies to improve the timing of notifications in activity recognition systems. By providing users with alternative notification times and considering the probability of activity occurrence, we intend to optimize the user experience and ensure that important activities are not overlooked or forgotten.

Our approach is based on a machine learning algorithm that uses historical data to predict future activities and then sends the appropriate notifications based on the probability and optimal timing. This approach also evaluates the impact of the notifications on user activity and provides feedback to the user, in addition to examining the effects of activities with low probability on forecasted activities. The purpose of this study is to address the following research questions:How can the integration of probabilistic and reinforcement learning techniques, specifically in the FaPTi (with probabilistic considerations) and FaTi (without probabilistic considerations) methods, optimize the timing of notifications for forecasted activities?What factors have an impact on an activity that has a low probability of needing to be carried out or not, and on needing a notification or not?What is the difference on user engagement between an activity and a an activity with a low-probability forecast?

Method comparisons were performed on the evaluation results of our proposed method with the baseline method for performance measures and probability levels for forecasted activities, and we also observed the characteristics of forecasted activities with low probability. The results show that the FAPTi and FaTi methods are superior to the baseline method, with FaPTi being better than FaTi. For performance measures, the percentage of positive responses and the response rate have a significant effect on the low probability of forecasted activity; and activities such as dressing up, daily chores, shopping, brushing teeth, taking a bath, working, snacking, having lunch, short time breaks, drinking, others, and walking have higher positive response values compared to response rates, while activities such as sleeping, learning at home, breakfast, dinner, biking, cleaning, and laundry have better response rates than positive response percentages.

This paper is organized as follows: Section 2 describes the relevant research on the notification system and forecasted activity notifications. This includes highlighting the distinctions between our work and related studies. In Section 3, we present the methodology employed in this research to develop the model and system architecture for data collection. Section 4 focuses on the participants involved in the study, the data collection procedures, the collected dataset, and the evaluation method. In Section 5, we present a comprehensive analysis of the obtained results, highlighting the key findings and insights derived from our study. Finally, Section 6 provides a conclusive summary of the paper, summarizing its main contributions and potential future directions for research in this field.

## 2. Related Work

### 2.1. Notification System

A notification system is a collection of protocols and procedures that involve both human and computer components, specifically designed to generate and send messages to individuals or groups of people. Notification systems play a vital role in our daily lives by keeping us informed and reminding us of important events and activities. The objective of these systems is to deliver timely and relevant notifications, thereby enhancing users’ productivity and overall experience. Extensive research has been conducted to explore various aspects of notification systems, including notification timing [33,34], content personalization [35,36], and user preferences [37].

In daily activities, notifications can be utilized as reminders for specific events [38,39], location-based alerts [40,41,42,43], and activity prompts [6,42,43,44]. The importance of notifications has led to an increasing demand for user attention on smartphones, as notifications serve as the primary means of conveying information to the intended recipients. However, untimely notifications can have adverse effects on individuals receiving them. Consequently, researchers have analyzed the impact of notification disruptions [45,46,47], such as their potential to disrupt work routines [46,48,49]. Consequently, notification optimization is considered necessary to ensure that important information can be received by users without hindering their performance, such as optimizing activity-based notifications based on the “filter-eligible” category to limit the maximum number of notifications [50], or optimizing notifications for multi-objective to exercise control over the volume of notifications sent using reinforcement learning [51]. However, in addition to notification volume, the timing of notification delivery, particularly for an activity with multiple notifications, poses another challenge that needs to be addressed in determining the most appropriate time to send notifications. This unresolved challenge serves as one of our motivations to perform notification optimization with multiple alternative timings to ensure that notifications capture users’ attention without causing disruption or annoyance.

As reminders, notifications can help provide important information about appointments [52], deadlines [53], or tasks that need to be completed [54]. With notifications, individuals can avoid delays or forgetting to carry out important activities. Additionally, notifications as reminders can also assist users in building positive habits. Systems can be used to remind users about physical exercise [55], health routines [56,57], or self-care activities [58] that need to be regularly performed. With the presence of notifications, individuals can stay consistent and organized in carrying out activities that support their well-being. For example, notifications can be used to remind individuals to exercise daily or take time for meditation and relaxation. Furthermore, notifications as reminders also aid in prioritizing and enhancing productivity. Systems can assist users in planning their schedules effectively and remind them of tasks that need to be prioritized. With timely notifications, users can avoid procrastination and manage their time efficiently. People can set their own reminders according to their preferences, but they cannot foresee upcoming events. Many things can occur beyond their scheduled activities due to busyness [10] or activity disruptions. Therefore, schedule changes [59] may happen, rendering pre-set reminders ineffective. Several studies have explored personalized notification strategies by employing techniques such as collaborative filtering [60], user profiling [61,62], a notification visualization system [63], and preference modeling [37]. By understanding users’ preferences and behavior, notification systems can deliver customized notifications that align with their interests and improve their engagement.

### 2.2. Forecasted Activity Notification

Forecasting is the process of predicting future outcomes based on present and historical data. Forecasting has been the subject of research in various fields, such as health informatics [14,15,16], human–computer interactions [17,18], and artificial intelligence [19,20]. This study aims to develop methods for predicting future activities, events, monitoring, or accidents. Several techniques have been proposed for forecasting, such as machine learning, probabilistic modeling, and rule-based systems [20]. The selection of a forecasting method is contingent on a number of variables, such as the context of the forecast, the availability and relevance of historical data, the desired degree of accuracy, the time period to be forecasted, and the business value of the forecast.

Forecasting enables researchers to assess potential risks and uncertainties. By analyzing historical patterns and trends, researchers can identify potential risks, such as a company’s response if market awareness is lower than expected [64], or challenges that may arise during their research projects. This allows for the development of contingency plans, risk mitigation, and necessary adjustments. The results of forecasting provide insights into potential changes, enabling information recipients to make better decisions. For example, flood forecasting for early preparedness [25,65], or flood recovery [66], is crucial for authorities and communities to implement short-term and long-term prevention measures. This is closely related to the delivery of information to users because if the information is not conveyed timely, such as the preparation of evacuation and rescue missions will be delayed. One way to deliver forecasted results is through notifications.

Previous studies have utilized the delivery of forecasted results through notifications in various domains. For example, messages sent via notifications to feed animals and manage dairy cows [22], notifying farmers about preventive actions and disinfectant spraying in agriculture [32], weather forecasting notifications [25,32,67,68], malaria notifications [30,31,69,70], honeybee activity with an alarm [71], bus delay notifications [72], and air quality notifications [73,74,75] as predictive information or utilize notifications to collect preliminary data on air quality [76]. However, most researchers primarily focused on consent to forecasting, considering notifications merely as information regarding the forecasted results. They did not pay much attention to whether the information delivered through notifications was received by users or not. Some researchers sent notifications after obtaining forecasted results or based on predefined thresholds [22,25,26,68], such as to inform about changes in honey bee hives based on temperature [71], while others scheduled separate delivery times for forecasted results [30,31]. Consequently, the delivery of notifications for forecasted results remains a challenge, particularly in the context of human daily activities.

In daily activities, the timing of notification delivery needs to be considered to avoid information overload or irrelevant information, ensuring that the activities are not missed, such as drinking enough water [77] and turning off the lights [78]. By delivering relevant and timely information through notifications to those involved, they can prepare and adapt to potential changes in the situation. However, forecasted activities often involve uncertain or disruptive aspects. Therefore, the timing of notification delivery for forecasted activities needs to be optimized.

In this work, we adopt a different approach to the delivery timing of forecast results via notifications, especially in the case of daily activities, by considering multiple alternative timings for notifications. In order to ensure the forecasting results information is received by the user at the appropriate time. We take into account the probability of whether the activity needs to be performed and whether notification is necessary for the forecasted activity. Ultimately, the available timing options are optimized using reinforcement learning to determine the best time for notification delivery.

## 3. Materials and Methods

The section on Materials and Methods provides an overview of the process used in this study to collect data and develop a model. This section is divided into two subsections: Model Development and System Architecture.

### 3.1. Model Development

In this section, we introduce the FaTi and FaPTi methods, designed to optimize forecasted activity notifications in daily life. The FaTi method utilizes the random forest algorithm for forecasting and reinforcement learning for time optimization. Figure 1 provides an overview of the FaTi method.

The FaPTi method incorporates the random forest algorithm for forecasting, Bayes’ theorem for probabilistic analysis, and reinforcement learning for time optimization. Figure 2 illustrates the overview of the FaPTi method.

The FaTi method in this study utilizes reinforcement learning to assign the optimal notification time for forecasted activities, while the FaPTi method incorporates probability calculations before applying reinforcement learning, allowing the determination of the likelihood that an activity must be performed and that a notification is required.

In this study, Bayes’ theorem as a formula to calculate conditional probabilities is used because this enables us to make informed decisions about whether to send a notification, in order to optimize the notification delivery for activities from forecasted that require user attention. In Bayes’ theorem, the conditional probability of an event (such as the probability of a user needing to perform an activity given that a notification is needed) is calculated using prior knowledge (such as the probability of a notification being needed) as well as the likelihood of observing the event (such as the probability of a user needing to do the activity). *P* (*NF*|*FA*) represents the conditional probability of a notification being needed given that a user needs to do the activity. This is calculated using Bayes’ theorem by multiplying the conditional probability of a user needing to do the activity given that a notification is needed (*P*(*FA*|*NF*)) with the prior probability of a user needing to carry out the activity (*P*(*NF*)) and dividing it by the prior probability of a notification being needed (*P*(*FA*)). Mathematically, the probability calculation is as follows: (1)$$P\left(NF|FA\right)=\frac{P(FA|NF)\times P(NF))}{P(FA)}$$

*NF:* User needs to complete the activity*FA:* User needs the notification

To determine the levels of activity that need to be carried out and need notifications, we set thresholds. The threshold value is obtained from calculating the mean value of probability for activities that need to be done and need notification, across all users. The resulting value of the probability is compared to the threshold value to determine the probability level of a user’s need to do the activity and the notification is needed. The probability level is configured such that if the probability value is below the threshold, the level is low. Despite this, the threshold is still quite high. The high and low levels of these probabilities will be put into a state of reinforcement learning as a product of time, input, and level (the FaPTi method).

In our proposed approach (FaTi and FaPTi methods), the user cannot set the time according to their wishes like most reminder systems [79], but the system will remind the user based on the estimated results and before the time of activity is executed because future events or disturbances are unpredictable, while the system will learn from the user’s activity behavior. We set the reminder time from the furthest distance to activity time. Our system will choose the action for the user, if the action is silent, then the user’s response is automatically no response; however, if the action is notified, then the user is prompted to respond by selecting one of the options that appear on the notification, such as NOW, LATER, or DISMISS. A NOW response indicates the user approves of the time and will prompt them to begin the activity immediately. A response from LATER indicates that the user appreciates the reminder time but will complete the activity later. A DISMISS response indicates that the user did not wish to be reminded at the current time. If the user does not select the option within five min due to negligence or another reason, our system determines that the user chose no response. After receiving a response from users, we optimize the timing of subsequent notifications based on each user’s response.

In the reinforcement learning part of the system, the system acts as the agent while the smartphone serves as the environment. The agent is responsible for observing the user’s context to determine whether to send a notification or remain silent. When the notification appears, the user can either NOW, LATER, or DISMISS it, and the response is then used by the agent to assign a reward. This process is aimed at maximizing the accumulated future reward discount, which is done using the Q-learning algorithm. The relationship between the agent and the environment is such that the agent makes observations to obtain a representation of the environment, which is referred to as the state. Based on the state, the agent takes actions that are determined by its policies. The environment then moves from the current state to the next state and returns a reward. The q-learning algorithm is a type of reinforcement learning that enables the agent to learn from the actions it takes. The algorithm updates the policy of the agent by evaluating the actions taken and their associated rewards. In this way, the agent can improve its performance over time. The method proposed in this paper aims to optimize notifications by providing the user with multiple alternative times for forecasted activities, thereby minimizing the chances of missing important activities.

In the reinforcement learning algorithm utilized by the FaTi method, the state is the product of time and input specified by the Equation (Equation 5)
(2)T={60,50,40,30,20,10,0}
(3)X={NOW,LATER,DISMISS}
(4)P={HIGH,LOW}
(5)S=T×X

In contrast, for the FaPTi method, we add the probability level to the state, transforming it into the product of time, input, and level defined in Equation (Equation 6).
(6)S=T×X×P

*T* means notification sending time, which is set before the activity time until it is equal to the activity time; the longest time is 60 min before the activity, and the shortest time is the same as the activity time. Each time has a difference of ten minutes. Table 1 shows the time features of our approach. *X* is the possible response that will be selected by the user, and we show the input features in Table 2. P is the probability level of the activity that needs to be completed or not and whether a notification is needed or not which has been defined previously, for the explanation of the level features we show in Table 3.

The reward function shows maps of the state–action pairs to return the corresponding reward value for a given state, defined in Equation (Equation 9)
(7)A={NOTIFY,SILENT}
(8)R={+1,−1,0}
(9)r:S×A→R

For the state transition function, we take the current state and action as input and return the next state. Mathematically, it can be represented as: (10)δ:S×A→S

For the policy function, we take the current state as input and return the action to be taken. Mathematically, it can be represented as: (11)π:S→A

Using the Q-Learning algorithm, the following steps are taken to calculate the q values to be updated in the q table:Initialize, for all $S_{t}$∈ S; $A_{t}$∈ A; t = 0.Start with $S_{0}$.At time step t, choose action $A_{t}$ = $argmax_{\left(a\in A\right)}$ Q($S_{t}$,a), ϵ-greedy is applied.Apply action $A_{t}$.Observe reward $R_{\left(t+1\right)}$ and move into the next state $S_{\left(t+1\right)}$.Update the Q-value function:
(12)$$Q\left(S_{t},A_{t}\right)\leftarrow Q\left(S_{t},A_{t}\right)+\alpha \left(R_{\left(t+1\right)}+\gamma max_{\left(a\in A\right)}Q\left(S_{\left(t+1\right)},a\right)-Q\left(S_{t},A_{t}\right)\right)$$Set t = t + 1 and repeat from step 3.

### 3.2. System Architecture

In our system, we use smartphones for data collection and apply the proposed method. Due to the expectation of self-labeling, participants are required to select an activity from a selection in our *FonLog* [80] app by clicking on a smartphone before and after performing the activity. Using this *FonLog* app, we have successfully recorded self-labeled data in a previous study [80]. On the notification side, we have two options for users: the notification response on the Android banner or the dialog form on *FonLog*. When a user responds to one of the forms, the other forms will also disappear. This is useful for user convenience in responding to notifications. Estimated results or forecasted activity will appear for four hours for each activity class, two hours before the activity time and two hours after the activity time. This is useful for facilitating participants in labeling. If the activity to be carried out is the same as the estimated result, they can click this menu, and then the start and finish time will be recorded automatically, but they can still modify it if there is a slight difference from the actual time. Figure 3 illustrates the user interface for *FonLog*.

For the method used, all processes are carried out on the server side, so it does not make the smartphone performance heavy. Due to this, an internet connection is required for incoming and outgoing data. Internet connectivity is essential for the device to function correctly. The data collected by devices need to be processed and analyzed in real time, which requires a stable and high-speed internet connection. The availability of internet connectivity is one of the crucial elements in implementing the Internet of Things (IoT) [81,82]. In daily activities, participants may be located anywhere, so we cannot guarantee that they will always be in a location with a reliable internet connection. Therefore, it is necessary to formulate a plan for internet connection issues. In our system, data are transmitted from the server to the user’s smartphone. When the user labels or responds to notifications, user-side data will be temporarily stored on the smartphone’s internal storage. The data will then be transferred to the cloud server when an Internet connection is established. This system also ensures that there is not an excessive amount of data on the local storage by automatically deleting data that have been effectively uploaded to the cloud. This cloud server performs data-processing tasks. Even though we use a cloud-based system, our system is not entirely reliant on the internet network; therefore, it can continue to function normally even if there are connection issues. In order to communicate, control, connect, and exchange data with other devices while still being connected to the internet, an object must be embedded with technology such as sensors and software.

## 4. Experimental Evaluation

The Experimental Evaluation section provides a summary of the data description and evaluation. The subsections of this section are Data Collection and Evaluation Method.

### 4.1. Data Collection

In this study, we used two datasets to evaluate our method, the EngagementService dataset [83] and data we collected in the field. Data were collected from 24 activities from six participants who used smartphone devices that had the *Fonlog* application installed. To determine the initial activity, nineteen activities were taken from the activity list to improve mobile health receptivity by employing environmental data and machine learning [84], then we distributed the initial activity list to the participants and asked them about their daily activities that were not included in the initial activity, and then we received five additional activities from the participants. We show the activity information in Table 4.

Forecasting is the process of predicting future activity based on past data, whereas, in reinforcement learning, agents learn from the environment, as described in Section 3.1. Therefore, in implementing the system, we first collect the initial data for three days. In this initial data collection, the proposed method has not been implemented. We set up the system so that on the first day, no reminders were sent to participants. Participants were instructed to select an activity from the list in *FonLog*, click the start recording button when they were about to perform the activity, and then click the stop recording button when they had completed the activity. On the second day, the reminders began to operate, not based on the forecasted activity but on the actual activity from the previous day. Notifications were generated randomly; to select an action in the form of notify or silent, seven time options will be chosen at random if the action is notified. For recording, we still followed the rules from the first day. On the third day, notifications for reminders were generated randomly based on the second day’s activities, and participants recorded the same activities as on the first and second days. From the information gathered in the initial data collection, activity forecasting, probabilistic, and reinforcement learning are implemented. We implemented the proposed method and the baseline method for nine days. At this stage, the reminder data no longer come from the previous day but from the forecasted activity results. The time required to send the notification was to be determined by time optimization utilizing reinforcement learning for the baseline method and probability calculations followed by time optimization utilizing reinforcement learning for the proposed method. We limit the notification time that appears on the participant’s smartphone to five minutes. If the user does not respond within five min, notifications on banners and dialog forms will automatically disappear. This is intended to prevent the user from receiving unnecessary notifications that could be distracting. Additionally, if there are too many unread notifications, it will be difficult for users to view them one by one.

In addition to conducting direct experiments, this paper also leverages data collected by previous researchers from the EngagementService dataset. The EngagementService dataset comprises various details such as CreationTime, AutoApprovalDelayInSeconds, Expiration, AcceptTime, SubmitTime, AutoApprovalTime, ApprovalTime, RejectionTime, RequesterFeedback, WorkTimeInSeconds, Input.content, Input.hour, Input.minute, Input.day, Input.motion, Input.location, Input.last_notification_time, Answer.sentiment. However, we only use information from Input.content, Input.hour, Input.minute, Input.day, Input.motion, Input.last_notification_time, and Answer.sentiment as user responses. Our simulations will center around seven activities, namely biking, bus, walking, train, stationary, driving, and running. These activities will be employed to assess the system’s performance and gather responses. To ensure diverse data, we will involve five subjects to extract responses from the available dataset. By selectively utilizing the relevant data fields and activities, we aim to derive valuable insights and evaluate the effectiveness of our system in user engagement analysis. By incorporating data from the EngagementService dataset and conducting our own experiments, we can gain a comprehensive understanding of the factors influencing user engagement across different activity types.

### 4.2. Evaluation Method

The comparison of the baseline, FaTi, and FaPTi methods was conducted to assess the feasibility of our proposed method. The baseline method involved random notification delivery at seven available time alternatives, ranging from sixty minutes before the forecasted activity time to the same time as the forecasted activity. Random notification delivery was chosen as the baseline because no previous method provided the optimization of notification timing for forecasted activities with multiple time options. This means that notification delivery for forecasted results in various fields of study [16,23,24,25,26] was solely based on a single-time forecasted result or employed different scheduling for notification delivery times [28,32]. We performed the simulation in which the activity and responses are collected from the EngagementService dataset [83] and the dataset from the implementation of the system as described in Section 3.2 and Section 4.1; this is intended to test our method’s efficacy in user-dependent scenarios. We compare the results of the FaTi and FaPTi methods using both datasets.

Three performance measures as criteria are defined in this paper: percentage of positive responses, user response rate, and response duration. With the percentage of positive responses, we can measure the proportion of user responses that are positive or favorable. This indicates how well the notification method is able to elicit a desired response from the users. A higher percentage of positive responses signifies that the users are actively engaging with the notifications and finding them useful. For example, if a notification is sent to remind users to complete a task, a high percentage of positive responses would indicate that the users are acknowledging the reminder and will take appropriate action. For user response rate, this metric represents the rate at which users respond to the notifications they receive. It measures the level of user engagement and interaction with the notifications. A higher response rate implies that the users are more actively involved and attentive to the notifications. It indicates the effectiveness of the notification method in capturing the users’ attention and motivating them to respond. A low response rate may suggest that the notifications are being ignored or overlooked by the users. Response duration is useful for measuring the duration of time it takes for users to respond to the notifications. It represents the speed or efficiency of the user’s response. A shorter response duration indicates that the users are promptly attending to the notifications and taking action. It reflects the effectiveness of the notification method in prompting timely responses from the users. On the other hand, a longer response duration may indicate delays or inefficiencies in the users’ engagement with the notifications.

By considering these three performance measures, a comprehensive evaluation of the notification methods can be obtained. Each metric provides unique insights into different aspects of the notification system’s performance. The percentage of positive responses assesses the impact and effectiveness of the notifications in eliciting desired user actions. The user response rate indicates the level of user engagement and attentiveness. The response duration measures the efficiency of the user’s response. Together, these metrics help in understanding the overall effectiveness, user engagement, and efficiency of the notification methods in optimizing the delivery of forecasted activities.

In this investigation, the positive response came from the NOW and LATER responses. The positive response is used for each dataset. The positive response is defined as follows: (13)$$\frac{RN+RL}{RN+RL+RD}\times 100$$

*RN*: the number of activities for which the response is now.*RL*: the number of activities for which the response is later.*RD*: the number of activities for which the response is dismiss.

Then, we determine the user response rate. The user response rate is defined as follows: (14)$$\frac{NR}{TN}\times 100$$

*NR*: the number of user responses to the notification.*TN*: total number of notifications sent.

Response duration refers to the length of time it takes for the user to respond to a particular stimulus or request. In this context, response duration can refer to the time it takes for a user to respond to a notification. This can include the time it takes to read and comprehend the message, as well as the time it takes to physically respond, such as by clicking a button. In the field of human–computer interactions, response duration is an important metric to consider when designing interfaces and systems. Long response durations can lead to frustration and decreased user satisfaction, while short response durations can enhance the user experience. The response duration is defined as follows: (15)duration=RT−NT

*RT*: the time when the user responds to a notification for each activity.*NT*: the length of time for which the notification appears on the user’s smartphone screen for each activity.

In the context of forecasting, it processes data from label information and time for each activity class, where time includes start and finish time. This allows for the provision of future activity information with the time duration to be generated. The label information is used as the target variable, while the start and finish time are used as features. The data will be split into three parts: training data, testing data, and validation data. Cross-validation is a technique used to evaluate the performance of the model, this is important for ensuring its generalizability to new data. In these approaches (FaTi and FaPTi methods), a leave-one-day-out cross-validation approach is used, where one day of data is set aside for validation, and the remaining data are used for training and testing. This approach ensures that the model is tested on data that it has not seen before, allowing for an accurate evaluation of its performance.

## 5. Results

In this section, we present the results of our study following our approach discussed above. The performance of our approach was evaluated based on the percentage of positive responses, user response rate, and response duration. To investigate the impact of integrating probabilistic and reinforcement learning techniques in optimizing the timing of notifications for forecasted activities, we illustrate the performance measurements for the EngagementService dataset in Figure 4, while Figure 5 presents the results of our data collection experiment dataset.

By comparing our approach with the baseline method (Figure 4 and Figure 5), we were able to assess the superiority of the FAPTi and FaTi methods in terms of their notification relevance and effectiveness. From the EngagementService dataset (Figure 4), the difference in the percentage of positive responses is not too significant, but the advantages of our approach are evident in the response rate and response duration. Meanwhile, from our experiment (Figure 5), the difference between all performance measurements was very clear, even for the response rate, the FaTi method was 25.26% superior to the baseline method, and the FaPTi method was 27.15% superior to the baseline method. The findings from this comparison serve as compelling evidence for the efficacy of our proposed approach in optimizing forecasted activity notifications. The inclusion of the baseline method in the study allowed us to establish a benchmark and provide a meaningful point of reference for evaluating the performance of our methods. By surpassing the performance of the baseline method, our approach demonstrates its potential to enhance the precision and reliability of notification time, ultimately leading to improved user experiences and outcomes.

To test the significant differences between methods on performance measures, we conducted statistical tests. We used *t*-tests to compare the performance between two different methods, aiding in understanding the variation or heterogeneity with a commonly set significance level for *t*-tests at 0.05 [85,86,87,88]. In this case, the first test was conducted between the FaTi method and the baseline. The results of the test showed a t-statistic of −0.26 with a *p*-value of 0.8036 for the positive response rate, −4.3 with a *p*-value of 0.0036 for the response rate, and 1.78 with a *p*-value of 0.1047 for the response duration. These results suggest that the response rate exhibits a statistically significant difference, while there is no statistically significant difference in the positive response rate and response duration between the baseline and FaTi methods. However, this does not imply that there is no difference at all between the two methods, but rather the observed differences are attributed to the variability among the sample data used in the testing. Next, the test was conducted between the FaPTi method and the baseline. We obtained a t-statistic of −1.31 for the positive response rate with a *p*-value of 0.2192, −3.78 for the response rate with a *p*-value of 0.0036, and 2.78 for the response duration with a *p*-value of 0.0196. This suggests that although there is a substantial difference in the percentage of positive responses between the baseline and FaPTi methods, the *t*-test results indicate that this difference is not statistically significant. However, for response rate and response duration, the high level of significance in the *t*-test indicates that the differences in both performance measures are statistically significant and can be considered stronger results in terms of significance compared to the difference in the percentage of positive responses.

In Figure 6 and Figure 7 we highlight the comparison of probability levels for forecasted activities to see the influence of each scoring criterion on the high and low probability. Through Figure 6 we observe the probability level for the forecasted activity from the simulation using the EngagementService dataset. The simulation results exhibit disparities compared to the baseline method, the percentage of positive responses has the lowest value of 27.88%, with the difference between high probability and low probability reaching 29.26%, and the difference in response rate is 11.55%, while the average response time has a difference of 3.85 min. On the other hand, through the FaTi and FaPTi method approaches, the low probability level still has a good response for each evaluation criteria. For example, in the FaPTi method, the difference between high and low probability levels of activities predicted is only 2.98%. Additionally, the response rate for low probabilities in the FaTi method is better than that for high probabilities, with a margin of 5.65%.

To further analyze our findings, we conducted an observation of the system implementation in the field, as depicted in Figure 7. In Figure 7, we highlight the comparison of probability levels for forecasted activities from our experiments. In the baseline method, there is a noticeable gap between high- and low-probability activity types in terms of performance measures. In particular, the performance measures for low-probability activities show a significant decrease compared to high-probability activities, with 25.96% for the percentage of positive responses and 38.21% for the response rate, while the mean response time is 1.44 min. However, when using the FaTi method, we observed an improvement in the performance measures for low-probability activities. Surprisingly, the performance measures for activities with high probabilities showed a decrease in this case, with a decrease of 13.52% for the percentage of positive responses and 4.01% for the response rate. On the other hand, the FaPTi method yielded positive results by enhancing performance measures for both high and low probability of forecasted activities, indicating that high probabilities maintained good performance while enabling favorable responses for low probabilities as well. This indicates that FaPTi effectively addresses the observed gap between the baseline and FaTi approaches, leading to improved recognition and notification of activities across a wider range of forecasted activity. Through these observations, we can see that the implementation of methods can influence the low probability of forecasted activities. Furthermore, the percentage of positive responses and response rate also play a role in determining the low probability of forecasted activities, while response duration has a relatively minor impact.

Regarding our previous observations, the FaPTi method outperforms the baseline and FaTi methods in terms of performance evaluation and probability levels. Therefore, in this part, we focus on the FaPTi method to examine the differences among forecasted activities with low probabilities, as illustrated in Figure 8.

Each activity exhibits distinct characteristics. Activities such as dressing up, daily chores, shopping, brushing teeth, taking a bath, working, snacking, lunch, short time breaks, drinking, other, and walking have higher positive response values compared to response rates. On the other hand, activities such as sleeping, learning at home, breakfast, dinner, biking, cleaning, and laundry have better response rates than positive response percentages. For activities like daily chores, shopping, snacking, short time breaks, other, and walking, users have higher positive response values but very low response rates. This indicates that for these activities, users choose to respond to notifications at their preferred times and ignore notifications they do not find appealing. Ignoring notifications implies that users have responded at the expected time and consider responding at a different time unnecessary. Activities like sleeping and breakfast have relatively high response rates but low positive responses, suggesting that users do not prefer the notification timing for these forecasted activities. For activities such as dressing up, taking a bath, dinner, biking, cleaning, and laundry, although some activities have higher positive response percentages than response rates, both the positive response percentages and response rates are relatively low, with no positive responses recorded for cleaning and laundry. This suggests that users do not have much time to respond to notifications for these activities. On the other hand, activities such as brushing teeth, working, learning at home, lunch, and drinking exhibit good positive response percentages and response rates, despite being forecasted to have low probabilities.

On another note, the mean response duration had the longest value at 4 min, and the fastest response at 0.13 min. This indicates that the time limitation for notifications displayed on smartphones is still relevant to users. However, this may impact the low response rate, implying that after five minutes, the notification will not appear on the smartphone screen, leading the system to assume no response from the user. Nevertheless, this approach can reduce notification fatigue for users.

We present personal characteristics in Figure 9. User 5 exhibits a slightly better response rate than the positive response and nearly achieves a balance between response rate and positive response. Meanwhile, other users have higher positive responses compared to response rates. In the case of User 2, there is a significant gap between positive response and response, indicating that User 2 chooses to respond only at preferred times and disregards most responses at unfavorable times. User 6 demonstrates excellent responses to positive responses and response rate. In contrast, User 3 shows low responses to both positive responses and response rates.

In summary, the FaPTi method demonstrates varying response patterns for different activity types with low probabilities. Some activities show higher positive response percentages, while others have better response rates. The duration of responses tends to fall within a reasonable range, highlighting the relevance of the time limitation for notifications on smartphones. This finding underscores the importance of considering users’ preferences and behaviors when optimizing the timing of forecasted activity notifications.

## 6. Discussion

This study aims to optimize the timing of notifications for forecasted activities by considering multiple alternative time options. We proposed two approaches as our contributions to this paper, the FaTI and FaPTi methods. The effectiveness and relevance of notifications were measured through three performance evaluations: the percentage of positive responses, response rate, and response duration, with two different datasets (EngagementService and our experiment) as explained in Section 4.1. To assess the efficacy of our proposed approaches in optimizing the notification of forecasted activity, we compared some metrics in FaTi and FaPTi with the baseline method.

By evaluating using both datasets, we addressed our first research question. Regarding the results of the comparison of methods described in Section 5, notable disparities were observed in response duration between our proposed approaches and the baseline method in the EngagementService dataset, but there were no substantial differences in terms of the percentage of positive responses and response rate. However, in the field experiment, clear distinctions were observed in all three performance evaluations. Both of our proposed approaches outperformed the baseline method, with a percentage of positive responses of 3.29% and a response rate of 25.26% for the FaTi method, and a percentage of positive responses of 18.38% and a response rate of 27.15% for the FaPTi method. However, the response durations did not exhibit notable distinctions, with 0.93 min for the FaTi method and 1.17 min for the FaPTi method. Overall, the FaPTi method exhibited superiority over the FaTi method and the baseline method across all performance measurements and datasets. This indicates that the comparison between our proposed approaches and the baseline method not only validates the effectiveness of our methods but also underscores the importance of integrating advanced techniques such as probabilistic and reinforcement learning in notification optimization for forecasted activity.

In response to the second research question, our study examined the comparison of probability levels of forecasted activity in both datasets to understand the factors influencing the low probability of activities needing to be done or not, and the need for notifications. Our findings revealed that the percentage of positive responses and response rate played a role in determining the likelihood of low-probability activities, whereas response duration did not have a substantial impact. Furthermore, our observations demonstrated that the application of the methods could also influence the low probability of forecasted activities, as seen in the substantial disparity between high and low probabilities when applying the baseline method. Additionally, in Figure 8 we can also see that low probability can affect the low response rate or percentage of positive responses. However, by selecting the appropriate timing for sending notifications to users, they still have the opportunity to respond to notifications at their preferred time, even though the busyness of future activities is unpredictable. This implies that a low probability for activities that need to be completed or not, and need notifications or not, does not imply complete neglect of reminding users, as these activities may not be necessary for the current day but will be needed in the future.

In response to the last research question, our case study showed the observed results of each activity for a low probability of activity predicted by the FaPTi method for our experiment. The intention behind this selection was that the FaPTi method had proven to be superior to FaTi and the baseline method. Additionally, our choice of a dataset for the experiment was motivated by the fact that individuals have varying levels of busyness each day, and daily activities may not be performed every day. Therefore, this aspect became our focus in analyzing field data. Some activities showed a high percentage of positive responses, such as daily chores, shopping, snacking, short time breaks, others, and walking. This indicates that users require reminders for these activities despite the low probability of forecasted activities. However, they are not always in front of their smartphone screens, resulting in the dismissal of notifications at other times when they are about to engage in or are currently performing activities. On the other hand, activities such as brushing teeth, working, learning at home, lunch, and drinking exhibited similar levels of positive response percentage and response rate, indicating that the system effectively learns and understands user behavior for these activities. In contrast, for activities such as cleaning and laundry, the system still needs to understand user behavior to determine the optimal notification delivery time due to the low percentage of positive responses and response rate. Thus, this study was able to demonstrate the characteristics of activities in user engagement with a low probability of forecasted activities.

In this work, we are unaware of the number of notifications users receive from various applications on their smartphones because the influx of notifications can lead users to ignore them more frequently. Although we have limited the appearance time of notifications on users’ smartphone screens to avoid notification overload, we cannot control notifications from other applications. Examining the number of notifications that appear on smartphones at specific times can provide better insights to enhance user engagement.

We hope that our proposed method and research findings will prevent individuals from missing out on future activities. Our system is designed to be dynamic, so if users can respond to notifications correctly, the timing will be optimized to facilitate users’ self-improvement and achieve a balance between important tasks and planned activities. Furthermore, in the future, this information contributes to improving data collection in research involving human activities.

## 7. Conclusions

In this paper, we propose methods for notification optimization by providing multiple alternative times for forecasted activity. Our proposed method optimizes notifications in a more effective manner. By incorporating the probability level of activity that needs to be done and needs notification into the state, we achieve a better performance measure than the baseline. Notifications are sent as reminders for upcoming activities, ensuring that individuals with prospective memory issues or busy schedules do not miss important tasks due to a lack of timely reminders. With timely notifications for future activities, users can better plan their tasks and improve productivity. In situations where unexpected activities arise, users can manage their activities more effectively and allocate the necessary time to complete urgent tasks. This helps avoid time wastage and allows users to stay focused on essential activities. Furthermore, by receiving relevant notifications, users feel informed and actively engaged in their activities. They can feel supported and assisted in their daily activities, regardless of whether the activities are planned or unplanned. This can enhance user satisfaction with the user experience of applications or systems that provide effective notifications.

The main contribution of this paper is providing time optimization to send notifications for forecasted activity through two approaches: the FaTi and FaPTi methods. We offer multiple time options for each forecasted activity. The FaTi method contributes to optimizing the time among the available options directly for the forecasted activities. The FaPTi method contributes by considering the probability of activity being necessary and requiring a notification based on the predicted activities before the notification timing optimization process takes place. We incorporate the state as a combination of time, the possible response of the user, and the probability level for an activity that needs to be done and needs notification, allowing the system to notify the user before the forecasted activity time. With the FaTi and FaPTi methods, it is possible to observe user responses over time and predict future timing and actions. By harmonizing exploration and exploitation, reinforcement learning can optimize the time of each user response.

Our proposed methods, FaTi and FaPTi, contribute to the field of activity recognition by offering innovative techniques to enhance notification timing, ultimately improving the usability and effectiveness of activity recognition systems.

Future work will focus on addressing activities with more complex problems such as in healthcare facilities, as the activities of medical staff, such as nurses, can vary on a daily basis. Additionally, the challenges of shift scheduling, including morning, afternoon, evening, and night shifts, pose further complications for individual scheduling. The varying shift assignments can disrupt their routine activities, adding to the complexity of the problem. To address these challenges, a larger dataset will be required, while ensuring that the data collection process does not interfere with the performance of medical staff. 

## Figures and Tables

**Figure 1 sensors-23-06510-f001:**
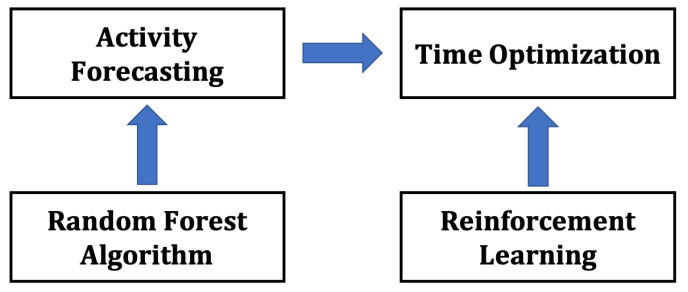
Proposed method (FaTi) overview. Notification optimization for forecasted activity with reinforcement learning.

**Figure 2 sensors-23-06510-f002:**
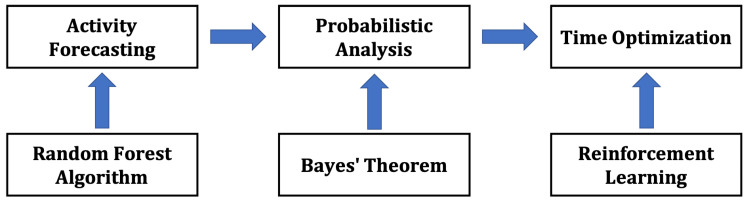
Proposed method (FaPTi) overview. Notification optimization for forecasted activity with probabilistic and reinforcement learning.

**Figure 3 sensors-23-06510-f003:**
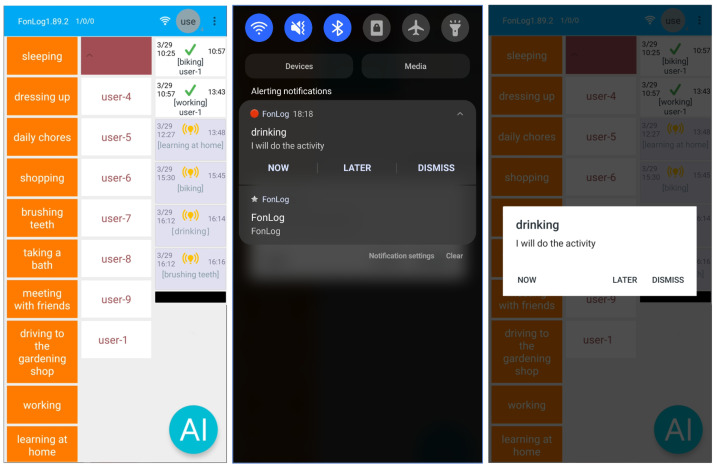
*FonLog* user interface.

**Figure 4 sensors-23-06510-f004:**
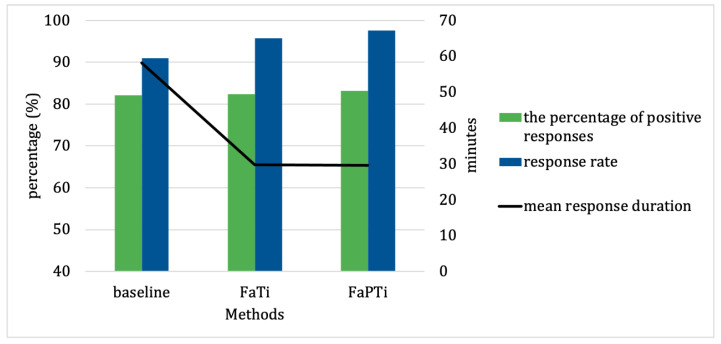
The Performance Measures Comparison From EngagementService Dataset. Positive response and response rate are displayed in percentage (%). Response duration is displayed in minutes.

**Figure 5 sensors-23-06510-f005:**
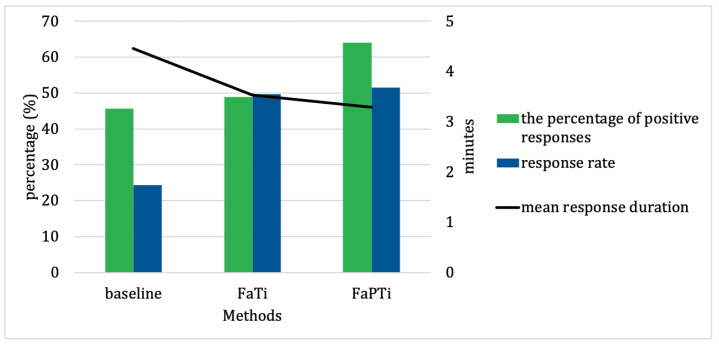
The Performance Measures Comparison From Our Experiment. Positive response and response rate are displayed in percentage (%). Response duration is displayed in minutes. We set the duration for notifications to appear on the smartphone screen to 5 min.

**Figure 6 sensors-23-06510-f006:**
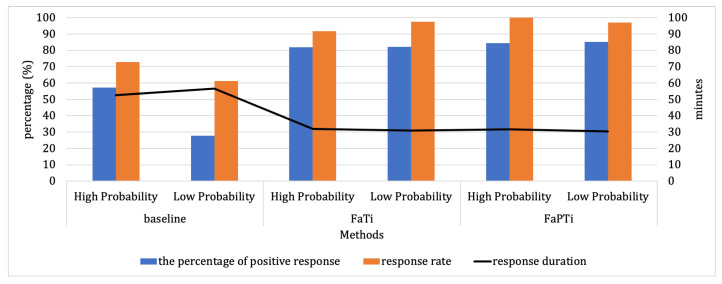
Probability Levels Comparison of Forecasted Activity From EngagementService Dataset. The lowest low probability value is found in the baseline method with a significant gap in the percentage of positive responses and response rate. In this dataset, there is no time limit on the display of notifications.

**Figure 7 sensors-23-06510-f007:**
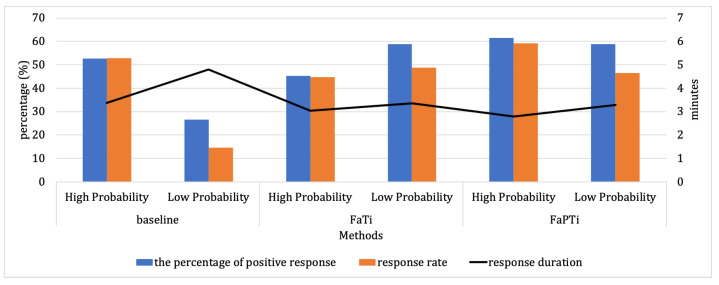
Probability Levels Comparison of Forecasted Activity From Our Experiment. The lowest low probability value is found in the baseline method. The FaPTi method is superior to the baseline and FaTi methods.

**Figure 8 sensors-23-06510-f008:**
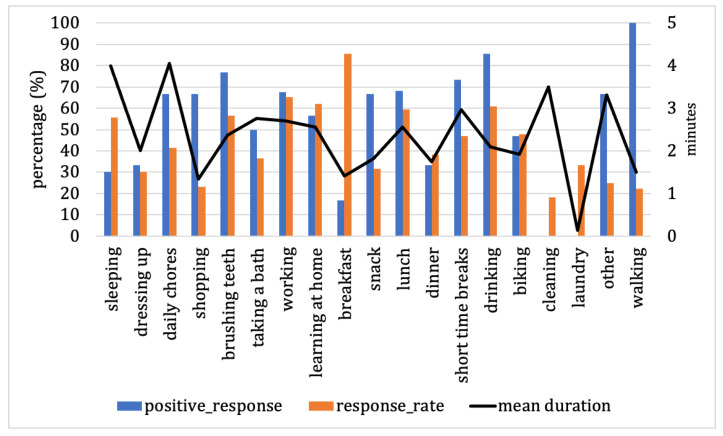
Characteristics of forecasted activity for low probability from our experimental dataset with the FaPTi method.

**Figure 9 sensors-23-06510-f009:**
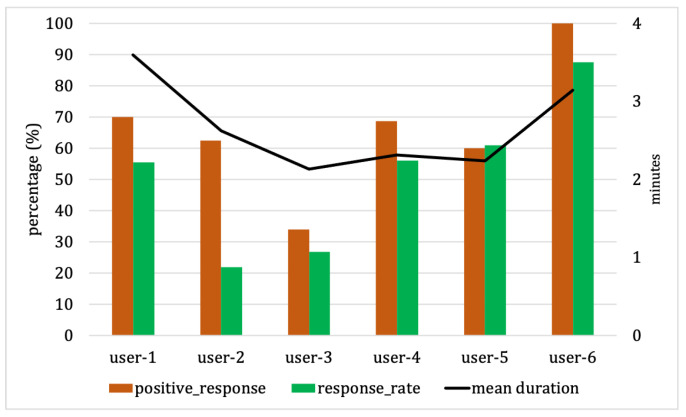
Personal characteristics for low probability from our experimental dataset with the FaPTi method.

**Table 1 sensors-23-06510-t001:** Time feature.

Time (T)	Description
60	60 min before the activity time
50	50 min before the activity time
40	40 min before the activity time
30	30 min before the activity time
20	20 min before the activity time
10	10 min before the activity time
0	the activity time

**Table 2 sensors-23-06510-t002:** Input feature.

Input (X)	Description
NOW	Users like notifications at this time, and they will complete the activity now.
LATER	Users like notifications at this time, but they will complete this activity later.
DISMISS	Users do not like notifications at this time, regardless of whether they will complete this activity or not.

**Table 3 sensors-23-06510-t003:** Level feature.

Level (P)	Description
high	the probability value is higher than the threshold
low	the probability value is lower than the threshold

**Table 4 sensors-23-06510-t004:** Activity types in the experiment.

Activity Name	Source
preparing to go to bed	[84]
sleeping	[84]
dressing up	[84]
daily chores	[84]
shopping	[84]
brushing teeth	[84]
taking a bath	[84]
meeting with friends	[84]
driving to the gardening shop	[84]
working	[84]
learning at home	[84]
mini-job	[84]
breakfast	[84]
snack	[84]
lunch	[84]
dinner	[84]
short time breaks	[84]
drinking	[84]
biking	Survey from the participants
washing dishes	Survey from the participants
cleaning	Survey from the participants
laundry	Survey from the participants
walking	Survey from the participants
other	[84]

## Data Availability

The data presented in this study are available on request from the corresponding author. The public dataset used is available online at: https://github.com/nesl/EngagementService (accessed on 17 February 2023).

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
