# Peer review of "Optimizing Forecasted Activity Notifications with Reinforcement Learning"

_sensors, 2023, doi:10.3390/s23146510_

Round 1

Reviewer 1 Report

The authors present two approaches (FaTi and FAPTi) aimed at improving the notification time in activity recognition system. The methods provide optimization for forecasted activity with reinforcement learning, sending notifications times in advance. According to the authors, FAPTi and FaTi methods are superior to the baseline method when comparing the performance based on the percentage of positive responses, user response rate, and response duration.

The paper should be accepted after a minor revision.

Comments:

1)      The paper provides a novel contribution on activity-based notification algorithms by providing new methods based on machine learning algorithms.

2)      The paper is well-written; however, it is recommended to be reviewed by an English expert for correcting minor errors if necessary.

3)      The small sample size of six participants may prevent the findings from being extrapolated.

4)      I recommend the authors to check if the statistical differences between the methods are significant through statistical tests.

5)      The authors may check if the following papers need to be included in the literature review:

Prabhakar, P., Yuan, Y., Yang, G., Sun, W., & Muralidharan, A. (2022, August). Multi-objective Optimization of Notifications Using Offline Reinforcement Learning. In Proceedings of the 28th ACM SIGKDD Conference on Knowledge Discovery and Data Mining (pp. 3752-3760).

Gao, Y., Gupta, V., Yan, J., Shi, C., Tao, Z., Xiao, P. J., ... & Chatterjee, S. (2018, July). Near real-time optimization of activity-based notifications. In Proceedings of the 24th ACM SIGKDD International Conference on Knowledge Discovery & Data Mining (pp. 283-292).

The paper is well-written; however, it is recommended to be reviewed by an English expert for correcting minor errors if necessary.

Author Response

Dear Reviewer,

Thank you for your letter and the opportunity to revise our paper on "Optimizing Forecasted Activity Notifications with Reinforcement Learning". The suggestions offered by the reviewers have been immensely helpful, and we also appreciate your insightful comments on revising the paper.
The changes are marked in blue in the paper, and we will also respond to comments based on the points in the description below:

Point 1: The paper provides a novel contribution on activity-based notification algorithms by providing new methods based on machine learning algorithms. 

Response 1: Thank you for recognizing the novelty of our paper's contribution. We appreciate your feedback and acknowledgement of our paper.

Point 2: The paper is well-written; however, it is recommended to be reviewed by an English expert for correcting minor errors if necessary.

Response 2: Thank you for your positive feedback on the overall quality of our paper. We have thoroughly revised the syntax and grammar of the sentences to ensure clarity throughout the paper.

Point 3: The small sample size of six participants may prevent the findings from being extrapolated. 

Response 3: We agree that the small sample size may prevent the findings from being extrapolated. In Section 5, specifically starting at line 577, we provide a detailed description of the characteristics of the individuals involved in the notification optimization for forecasted activities. These findings can be considered reliable, particularly in the context of personalized approaches where even a single subject can provide valuable insights. However, for generalization, we cannot be certain whether six participants are sufficient or not, as there is no fixed or universal number that determines the feasibility of extrapolation.

Point 4: I recommend the authors to check if the statistical differences between the methods are significant through statistical tests.

Response 4: Thank you for your recommendation. We agree with your suggestion, and we have incorporated statistical tests to assess the significance of the differences between the methods. In Section 5, starting from line 494, we have included the results of t-tests to evaluate the statistical differences between the methods. These tests provide valuable insights into the comparative performance of the approaches and further strengthen the validity of our findings.

Point 5: The authors may check if the following papers need to be included in the literature review:
- Prabhakar, P., Yuan, Y., Yang, G., Sun, W., & Muralidharan, A. (2022, August). Multi-objective Optimization of Notifications Using Offline Reinforcement Learning. In Proceedings of the 28th ACM SIGKDD Conference on Knowledge Discovery and Data Mining (pp. 3752-3760).
- Gao, Y., Gupta, V., Yan, J., Shi, C., Tao, Z., Xiao, P. J., ... & Chatterjee, S. (2018, July). Near real-time optimization of activity-based notifications. In Proceedings of the 24th ACM SIGKDD International Conference on Knowledge Discovery & Data Mining (pp. 283-292). 

Response 5: We have thoroughly reviewed the papers you recommended, and we have included them in our literature review. In Section 2, specifically in lines 138-148, we have discussed these papers and highlighted the distinctions between our research and the findings presented in these works.

Reviewer 2 Report

Authors proposed a notification optimization method by providing multiple alternative 1 times for forecasted activities with and without probabilistic considerations for the activity. However, there are still some concerns that need to be addressed.

1.The writing of the manuscript needs to be further polished. The authors should carefully check the manuscript for writing and conceptual description.

2. The paper’s organization should be added in the last part of the introduction.

3. Authors should highlight contributions in the introduction and conclusion sections. 

4. The authors should report the performance of other methods, so that show the advantages of the proposd method.

5. Reinforcement learning is widely used in various  tasks, please cite the fellowing works.

[1] Safe learning in robotics: From learning-based control to safe reinforcement learning

[2] Reasoning Structural Relation for Occlusion-Robust Facial Landmark Localization 

[3] Stable-baselines3: Reliable reinforcement learning implementations

The writing of the manuscript needs to be further polished. The authors should carefully check the manuscript for writing and conceptual description.

Author Response

Dear Reviewer,

Thank you for your letter and the opportunity to revise our paper on "Optimizing Forecasted Activity Notifications with Reinforcement Learning". The suggestions offered by the reviewers have been immensely helpful, and we also appreciate your insightful comments on revising the paper.
The changes are marked in blue in the paper, and we will also respond to comments based on the points in the description below:

Point 1: The writing of the manuscript needs to be further polished. The authors should carefully check the manuscript for writing and conceptual description. 

Response 1: Thank you for your feedback on the writing quality and conceptual description of our manuscript. We have taken your comments into serious consideration and conducted a thorough revision of the manuscript. Specifically, we have meticulously reviewed and polished the syntax and grammar of the sentences to ensure clarity and coherence throughout the paper. Additionally, we have add descriptions for the conceptual in the Introduction by providing further explanations, such as in lines 56 and 72, to enhance the overall understanding of the topic.

Point 2: The paper’s organization should be added in the last part of the introduction.

Response 2: We agree with your recommendation, and we have incorporated the paper's organization in the concluding part of the Introduction section. Starting from line 113, readers will find a clear outline of the paper's structure, which will guide them through the subsequent sections and provide a comprehensive overview of the content.

Point 3: Authors should highlight contributions in the introduction and conclusion sections. 

Response 3: Thank you for your feedback regarding highlighting the contributions in the introduction and conclusion sections. We have already written the contributions in both the introduction and conclusion parts of the paper. However, in order to further clarify our contributions, we have made modifications to specific sentences, such as in line 72 of the Introduction section. Additionally, in the conclusion, we have revised the opening sentences of the first and second paragraphs to ensure that the contributions are explicitly and clearly stated

Point 4: The authors should report the performance of other methods, so that show the advantages of the proposed method.

Response 4: Thank you for your comment regarding reporting the performance of other methods. We appreciate your suggestion, but in our specific research context, we did not come across any existing methods that directly address the objective of optimizing notification with several alternative time for forecasted activities. While we have reviewed similar related work, none of them specifically focus on the same approach as ours. As a result, it was not feasible to provide a direct comparison with other methods. However, we have thoroughly discussed and highlighted the unique advantages and contributions of our proposed method in the manuscript. We believe that our approach fills an important gap in the existing literature and offers novel insights and improvements in the optimization of timing for forecasted activities.

Point 5: Reinforcement learning is widely used in various tasks, please cite the fellowing works.
[1] Safe learning in robotics: From learning-based control to safe reinforcement learning
[2] Reasoning Structural Relation for Occlusion-Robust Facial Landmark Localization 
[3] Stable-baselines3: Reliable reinforcement learning implementations 

Response 5: We agree with you that reinforcement learning is widely utilized in various tasks. We have thoroughly reviewed the papers you mentioned. After careful consideration, we found that the objectives and approaches presented in those papers are significantly different from our work. While these papers incorporate reinforcement learning, their focus and application contexts are distinct from our objective and approach.
[1] focuses on learning-based control and safe reinforcement learning in robotics systems to address uncertainty and safety constraints.
[2] specifically addresses occlusion-robust facial landmark localization using a structural relation network (SRN) in video analysis.
[3] Stable-Baselines3 provides an open-source implementation of deep reinforcement learning algorithms as a Python library.
Although these papers explore aspects related to reinforcement learning, they do not directly align with the specific objectives and approaches of our work. Nonetheless, we appreciate your suggestion and the opportunity to review these papers. Their insights have contributed to our understanding of the broader applications of reinforcement learning.

Reviewer 3 Report

The authors of this paper proposes a time optimization in the delivery of notifications for predicted activities through two approaches: the FaTi and FaPTi methods. They offer multiple time options for each predicted activity. The FaTi method contributes to optimizing the time among the available options directly for the forecasted activities. The FaPTi method contributes by considering the probability of activity being necessary and requiring a notification based on the predicted activities before the notification timing optimization process takes place. The authors, last,  incorporate the state as a combination of time, the possible response of the user, and the probability level for an activity that needs to be done and needs notification, allowing the system to notify the user before the forecasted activity time.

Ηowever the presentation quality needs some improvement:

-Recent references after 2020 should be added as well as a comparison (via a table) of the recent bibliography with the proposed techniques.

-Ιn the subsection 3.1 there is no need to give such a detailed description to the idea of "Bayesian inference" which are widely known in the literature.

-Τhe bullets below relation (1) are wrong, the explanation of the terms (NF), (FA) was reserved.

-The definition "mean propability value" in line 232 is somewhat misleading.  Do you mean the mean value or the propability value (p-value)?

- Could the proposed system model be formalized  as a dynamic Bayesian network (DBN)? As a probabilistic graphical model, the DBN considers a set of variables and their conditional dependencies over adjacent time steps. In this way, it can be generated a stochastic simulator to make decisions based on both contextual and cognitive states sequentially.

- The relation (11) should be made more representative and replaced with π(s)=argmaxaQ(s,a)

Or a bit more formally:

π:SA=argmaxaA(s)Q(s,a)sS

argmaxaQ(s,a)  or  aA(s)S

The quality of English language is good.

Author Response

Dear Reviewer,

Thank you for your letter and the opportunity to revise our paper on "Optimizing Forecasted Activity Notifications with Reinforcement Learning". The suggestions offered by the reviewers have been immensely helpful, and we also appreciate your insightful comments on revising the paper.
The changes are marked in blue in the paper, and we will also respond to comments based on the points in the description below:

Point 1: Recent references after 2020 should be added as well as a comparison (via a table) of the recent bibliography with the proposed techniques. 

Response 1: We agree with your recommendation to add recent references after 2020. We have added some relevant references published after 2020, as in lines 141, 143, 165, 166, and 191–202. While we have reviewed similar related work, it is important to note that these studies address similar aspects but do not directly align with our specific approach or problem. Therefore, it would be difficult for us to provide a comprehensive comparison in a table format. However, we have provided detailed explanations highlighting the differences between our work and related studies, as presented in lines 138–148, 202-203, and 211–217. These explanations elucidate the distinct contributions and approaches in a comprehensive manner.

Point 2: Ιn the subsection 3.1 there is no need to give such a detailed description to the idea of "Bayesian inference" which are widely known in the literature.

Response 2: We agree with your feedback regarding the detailed description of "Bayesian inference" in subsection 3.1. As it is a well-known concept in the literature, we have revised and reduced the explanation in that section accordingly. We believe that this modification will improve the clarity and conciseness of the manuscript while ensuring that the essential information is still provided.

Point 3: Τhe bullets below relation (1) are wrong, the explanation of the terms (NF), (FA) was reserved. 

Response 3: We apologize for the confusion and thank you for pointing out the error. Upon reviewing the manuscript, we have identified that the mistake was not in the bullets themselves but in the preceding explanation. We have addressed and corrected this error in the revised version of the paper. The necessary modifications have been made in lines 242-245 to ensure the clearly the description.

Point 4: The definition "mean propability value" in line 232 is somewhat misleading. Do you mean the mean value or the propability value (p-value)?

Response 4: We appreciate the reviewer's attention to detail and thank them for bringing this to our attention. You are correct, the term "mean probability value" was somewhat misleading in its original formulation. The intended meaning was "the mean value of probability." We have carefully revised the sentence in line 250 to remove any ambiguity and provide a clear description. Please note that the line number mentioned (232) has changed due to revisions made in the preceding lines.

Point 5: Could the proposed system model be formalized as a dynamic Bayesian network (DBN)? As a probabilistic graphical model, the DBN considers a set of variables and their conditional dependencies over adjacent time steps. In this way, it can be generated a stochastic simulator to make decisions based on both contextual and cognitive states sequentially.

Response 5: Thank you for your insightful comment regarding the potential formalization of our proposed system model. We understand that a dynamic Bayesian network (DBN) is capable of capturing temporal dependencies and dynamic behavior in probabilistic systems. The graphical representation in DBN visually illustrates the conditional dependencies among variables. However, our proposed system differs in approach or methodology as we utilize reinforcement learning, allowing for dynamic decision-making based on rewards. This is a key distinction from a dynamic Bayesian network approach, as reinforcement learning enables dynamic learning and adaptation through rewards whereas DBN cannot learn dynamically with rewards, we want to do it in a dynamic way which means we aim to incorporate dynamic learning in our approach. We believe that our proposed system model contributes valuable insights to this research area, opening avenues for further work on combining reinforcement learning with dynamic Bayesian networks, such as use some state transition function using Bayesian networks. However, before evaluating the basic reinforcement learning approach presented in this paper, we cannot go for a dynamic Bayesian reinforcement learning network. To use a dynamic Bayesian network for reinforcement learning requires further considerations such as conditional variable, before integrating reinforcement learning with a dynamic Bayesian network.

Point 6: The relation (11) should be made more representative and replaced with π(s)=argmaxaQ(s,a)
Or 
a bit more formally: π:S→A=argmaxa∈A(s)Q(s,a)∀s∈S 
argmaxaQ(s,a) or a∈A(s)S

Response 6: We agree that this modification will enhance the clarity and precision of the equation. We have made the necessary revisions to relation (11) in the paper.